

# Insights into dog owner perspectives on risks, benefits, and nutritional value of raw diets compared to commercial cooked diets

Alysia Empert-Gallegos*, Sally Hill* and Philippa S. Yam

School of Veterinary Medicine, College of Medical, Veterinary, and Life Sciences (MVLS),
University of Glasgow, Glasgow, UK
* These authors contributed equally to this work.

## ABSTRACT

**Background:** The practice of feeding a raw meat-based diet (RMBD) to dogs is a topic of increasing interest to owners and veterinary professionals alike. Despite the research around the practice, particularly about the risk of nutritional imbalances and microbial contamination, an increasing number of dog owners are adopting a raw diet for their dogs. This study expands the research into owner motivations for feeding RMBDs and cooked diets and asks them their opinions about risk and nutritional value.

**Methods:** An anonymized, online, internationally accessible questionnaire was developed to ascertain owner perspectives on the risks, benefits, and nutritional value of commercially prepared and homemade RMBDs as compared with commercially prepared cooked diets (CCDs).

**Results:** The questionnaire was completed by 419 dog owners of diverse backgrounds across the world. Of the participants, 25.3% fed RMBDs. Just over 70.0% of all participants had spoken to their veterinarian about their dog's nutrition. Owners who fed RMBDs ranked their veterinarian's knowledge lower and their own knowledge of canine nutrition higher than owners who fed CCDs. They rated commercial and homemade RMBDs as highly nutritious 83.5% and 73.6% of the time, respectively, while only 12.5% rated CCDs as highly nutritious. Owners who fed CCDs ranked RMBDs as highly nutritious less often, but also only ranked CCDs as highly nutritious 52.7% of the time. All participants agreed that CCDs were low risk to human health. Owners who fed RMBDs ranked raw diets as highly risky to human or dog health under 20.0% of the time but deemed CCDs risky to animal health over 65.0% of the time. When asked about benefits of raw diets, the most repeated words offered by owners were "health", "better", "coat" and "teeth". The most repeated risks presented were "bacteria", "nutrition", "risk" and "Salmonella". Owners who fed RMBDs tended to use vague terminology like "health" and "better" when asked why they fed a raw diet. Owners who did not feed RMBDs used more specific terminology like "expensive", "time" and "risk" when asked why they did not feed a raw diet. Overall, the two groups differed in their perceptions around RMBD and CCD feeding, which highlights the need for a better line of communication and education between veterinarians and owners.

Corresponding author
Sally Hill,
2217479p@student.gla.ac.uk

## INTRODUCTION

The practice of feeding a raw meat-based diet (RMBD) to dogs is a topic of increasing interest to owners and veterinary professionals alike, with Google searches for "raw dog food" quadrupling over the last 10 years (*Google Trend Data, 2020*). Estimates of the percentage of dog owners who feed RMBDs vary, but experts agree that the practice is increasing in both the United States and Europe (*Davies, Lawes & Wales, 2019*).

As the feeding of RMBDs has become more common there has been an increase in research on the safety of the practice, particularly the potential for nutritional imbalances and the microbiological risks associated with raw meat. Leading veterinary organizations, from the American Veterinary Medical Association to the World Small Animal Veterinary Association, discourage the feeding of RMBDs, but given its increasing popularity some owners are clearly not heeding their warnings (*American Veterinary Medical Association (AVMA), 2019*; *World Small Animal Veterinary Association (WSAVA), 2017*). Despite this increased popularity and interest in RMBDs, scant research has been published on dog owner opinions regarding this type of feeding and comparing it to the feeding of commercially prepared cooked diets (CCDs).

This study was undertaken with the aim to better understand dog owner perspectives on risks, benefits, and nutritional value of raw diets compared to cooked diets. We hypothesized that owners who feed primarily raw diets to their dogs will perceive the practice to pose less risk to both human and animal health than those who feed a non-raw diet. We also anticipated that those who feed raw diets will perceive them as more or equally nutritious as CCDs.

## MATERIALS AND METHODS

Approval to conduct the project was sought and granted by the University of Glasgow College of Medical, Veterinary & Life Sciences Ethics Committee for Non-Clinical Research Involving Human Participants (Approval Ref: 200180125).

An anonymized, online, internationally accessible, open questionnaire was developed using Google Forms. The questionnaire was tested against peers to assess usability, technical functionality, comprehension, cohesiveness, flow, and length before fielding the study. The questionnaire consisted of 11 sections. The first section included a description of the questionnaire including time estimate for completion, stated the terms of consent, and included contact information for the researchers. Any person over the age of 18 who was the primary owner or caregiver of a pet dog was invited to participate as the target population. Participation was completely voluntary and required participants to consent to continue with the questionnaire. Participants were not required to sign in or belong to any network in order to participate. No incentives for participation were offered. The second section collected demographics about the dog owner. Sections 3–10 included dichotomous, categorical, ordinal and free-text box questions. Sections 3–10 included 1–4 questions each, with an average of three questions per section. Section 11 consisted of a brief appreciation for their time and ended the questionnaire. Participants were allowed to skip any question or answer free-text questions with "NA" (not applicable) and they were able to navigate forwards and backwards before submitting the

questionnaire. Questions asked about general feeding trends and opinions, rather than asking for owners to respond about each pet they may own.

Results were automatically collected by the Google Forms software and were exported from the internet when the data collection was finished.

After obtaining permission from page administrators, a live, sharable link to the questionnaire was initially posted four times on dog-centric community pages on Facebook; Dogspotting (1.7 million members) and Dogspotting Society (974,000 members). Facebook users frequent these types of pages to view and share photos or videos of dogs and information about dog ownership. Two of the researchers also posted the link on their personal Facebook pages. It is known to the researchers that the survey was posted elsewhere on Facebook by individuals other than the researchers. The link remained live for 14 weeks during the summer of 2019.

Discrete data were compiled and analyzed using Microsoft Excel. Statistical significance was determined using chi-squared testing with a $p$-value of less than 0.001 unless specifically noted. When presenting percentages to compare cooked and raw feeding groups, summation of percentages is always 100%. Qualitative analysis of the fill-in data was undertaken using the RStudio text mining (tm) package and word cloud generator package (wordcloud) to discover relevant word frequencies and visually represent the data. Qualitative data were mined to exclude common English stop words and combine words with the same stem (e.g., risk and risky) as well as words with an identical meaning (e.g., stool, feces, faeces, poop, poo).

## RESULTS

### Demographics

Four hundred and nineteen people of variable age responded to the survey. Most respondents were female ($n = 393$, 93.6%), omnivorous ($n = 357$, 85.6%), and did not work as part of the animal industry or animal related field ($n = 306$, 73.2%). Only a minority of respondents lived in households with immunocompromised individuals ($n = 33$, 7.9%), pregnant women ($n = 10$, 2.4%) or children under 10 years of age ($n = 43$, 10.3%). Our survey participants hailed from 16 countries on five continents, with the United States ($n = 206$, 58.4%) and the United Kingdom ($n = 97$, 27.5%) as the most represented and second most represented countries, respectively (Table 1).

### Establishment of diet

When asked how owners established their dog's diet, 24.4% ($n = 101$) of respondents said they followed a recommendation from a veterinarian, veterinary nurse or veterinary technician. Information published online from a non-veterinary source ($n = 54$, 13.0%) and information published by a veterinarian or veterinary nutritionist ($n = 50$, 12.1%) were also popular choices. The remainder of respondents ($n = 209$, 50.5%) established their dog's diet in myriad other ways, like recommendations from friends and family, tradition (what they have always fed), recommendation of the breeder or rescue/shelter and recommendation from breed specific literature, among others (101 discrete answers were offered).

| Table 1 Demographic data for survey respondents. | | |
|---|---|---|
| **Variable** | **Number of respondents** | **%** |
| Total survey respondents | 419 | 100 |
| **Gender** | | |
| Apogender | 1 | 0.2 |
| Female | 393 | 93.6 |
| Male | 23 | 5.5 |
| Nonbinary | 1 | 0.2 |
| Prefer not to say | 2 | 0.5 |
| **Age (years)** | | |
| 18–24 | 60 | 14.3 |
| 25–35 | 146 | 34.8 |
| 36–45 | 60 | 14.3 |
| 46–55 | 71 | 16.9 |
| 56–65 | 64 | 15.3 |
| Over 65 | 18 | 4.3 |
| **Country of Residence** | | |
| Australia | 11 | 3.1 |
| Canada | 25 | 7.0 |
| New Zealand | 2 | 0.6 |
| Singapore | 2 | 0.6 |
| South Africa | 1 | 0.3 |
| Spain | 1 | 0.3 |
| Turkey | 1 | 0.3 |
| United Kingdom | 97 | 27.2 |
| United States | 206 | 57.9 |
| Other: One respondent from each country (China, France, Hong Kong, Italy, Malaysia, Norway, Portugal, South Africa, Spain, Turkey) | 10 | 2.8 |
| **Industry (Animal-related field?)** | | |
| Animal Industry (pet food, toys, products) | 12 | 2.9 |
| Animal Services (groomer, farrier, acupuncturist, kennel staff, etc.) | 37 | 8.9 |
| Breeder | 5 | 1.2 |
| No. Other profession not related to animal industry | 306 | 73.2 |
| Student of veterinary medicine/science/nursing | 34 | 8.1 |
| Veterinarian or Vet Nurse | 24 | 5.7 |
| **Owner's Dietary Preferences** | | |
| Omnivore (meat and plant-based diet) | 357 | 85.6 |
| Vegan | 9 | 2.2 |
| Vegetarian | 39 | 9.4 |
| Other | 12 | 2.9 |
| **Household with children under 10 years of age** | | |
| No | 375 | 89.7 |
| Yes | 43 | 10.3 |
| **Household with immunocompromised individual** | | |
| "I don't know" | 8 | 1.9 |
| No | 376 | 90.2 |
| Yes | 33 | 7.9 |
| **Household with pregnant individual** | | |
| "I don't know" | 1 | 0.2 |
| No | 408 | 97.4 |
| Yes | 10 | 2.4 |

Owners were asked what their dogs ate most of the time (their main diet). The majority of respondents fed their dog a CCD as the main diet, in the forms of dry/kibble or wet/canned/sachet ($n = 267$, 63.7%). Commercially prepared raw food ($n = 54$, 12.9%) was the next most popular choice, followed by a homemade raw diet ($n = 52$, 12.4%), a prescription diet for a medical condition ($n = 30$, 7.2%) and a homemade cooked diet ($n = 16$, 3.8%). For the purposes of this paper, owners who fed predominantly raw diets (as their main diet) in any form will herein be referred to as "raw feeders" while owners who fed predominantly cooked diets in any form will be referred to as "cooked feeders".

## Perceptions of diets

Respondents were asked to rate various diets on scales of 1 to 5 (where 1 is lowest and 5 is highest) in terms of perceived nutritional value, as well as risks to both human and dog health; the diets they rated were CCDs, commercially prepared RMBDs, and homemade RMBDs.

When asked about nutrition, raw feeders were more likely to rate RMBDs as highly nutritious (4 or 5 out of 5) than cooked feeders were ($p < 0.001$ for all comparisons made), and made only a small distinction between commercial or homemade RMBDs, with 83.5% ($n = 79$) of raw feeders rating commercial RMBDs as highly nutritious and 73.6% ($n = 78$) rating homemade RMBDs as highly nutritious. In contrast, cooked feeders made a greater distinction between commercial and homemade diets than between cooked and raw; 52.7% ($n = 164$) of cooked feeders rated CCDs as highly nutritious and 51.6% ($n = 119$) gave the same rating to commercial RMBDs. Only 35.7% ($n = 106$) of cooked feeders rated homemade RMBDs as highly nutritious, and only 12.5% ($n = 13$) of raw feeders rated CCDs as highly nutritious (Fig. 1).

When asked about risk, all participants rated CCDs as low risk to humans; there was no statistical difference between the groups ($p = 0.93$). However, cooked feeders perceived RMBDs as riskier to both their dog's health and human health than raw feeders did ($p < 0.001$ for all). Both groups viewed CCDs as riskier to dog health than to human health: while approximately 15.0% of both groups thought there was a high risk to human health (cooked feeders $n = 17$, raw feeders $n = 7$), there was a significant difference in perception of risk to dog health between groups. Forty-five cooked feeders (25.1%) rated CCDs as highly risky to dog health. The results from raw feeders were more extreme: 65.3% ($n = 54$) of them rated CCDs as highly risky to dog health (Fig. 1).

## Perceptions of nutritional knowledge

Two hundred and ninety-seven (70.9%) respondents had discussed nutrition with their veterinarian. When asked to rank how knowledgeable they felt their veterinarian was about nutrition where 1 was least knowledgeable and 5 was most knowledgeable, 59.9% ($n = 249$) of respondents gave their veterinarian a 4 or 5 out of 5. When asked to rank how knowledgeable owners felt they were about their dog's nutrition, 61.9% ($n = 260$) gave themselves 4 or 5 out of 5. Further analysis of this data shows a significant difference in responses given by raw feeders and cooked feeders. When asked to rank their own nutritional knowledge on a scale of 1 to 5, 65.2% ($n = 174$) of cooked feeders ranked
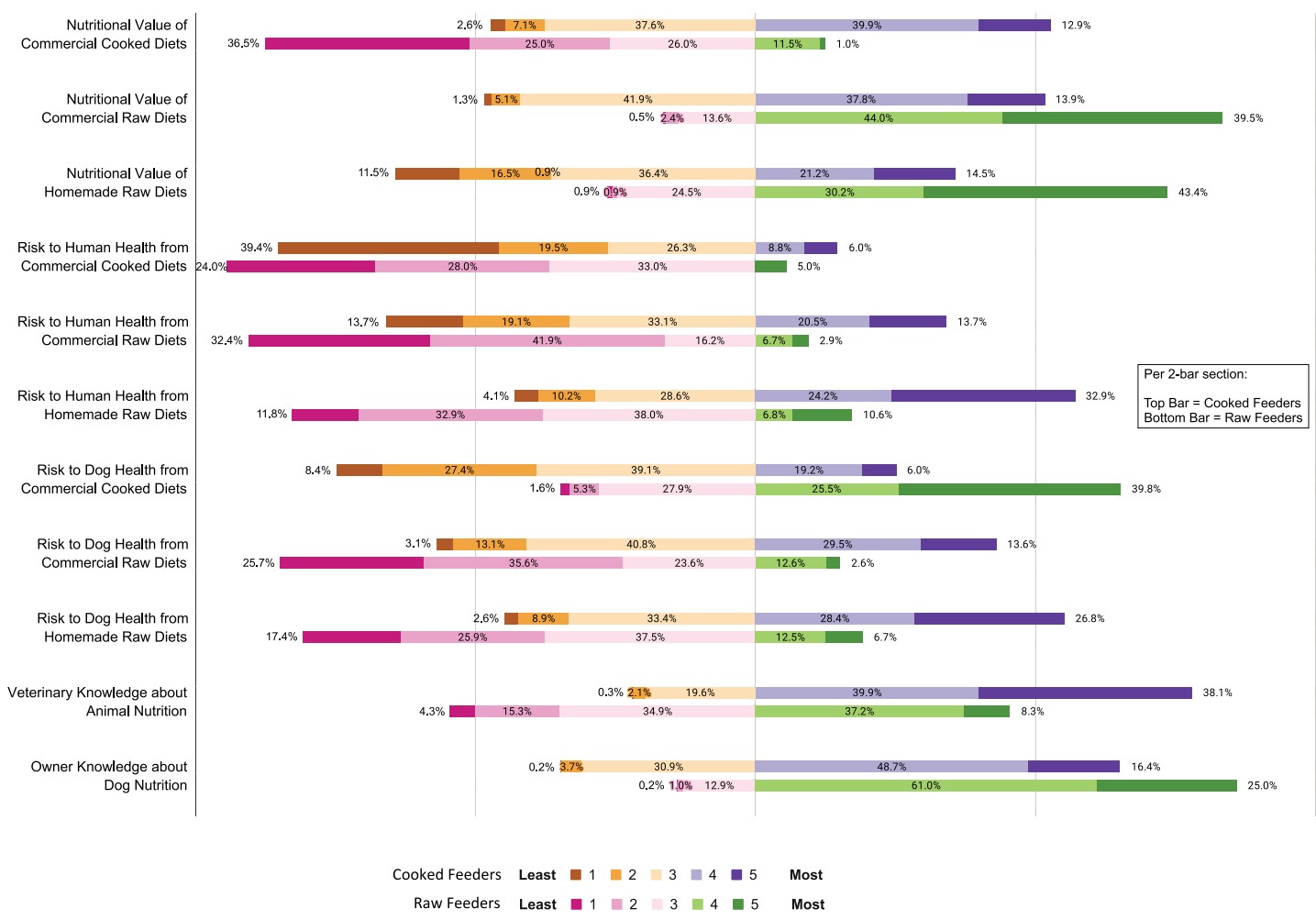

**Figure 1 Owner responses to rating questions.** Dog owner responses when asked to rate cooked diets or raw diets on three variables (nutritional value, risk to dog health, or risk to human health) as well as their own knowledge of nutrition and their veterinarian's knowledge of nutrition on a 5-point scale, where 1 was the least and 5 was the most of each variable, separated by owner's chosen diet (cooked feeders vs raw feeders).

themselves as 4 or 5 out of 5, while 86.0% ($n = 85$) of raw feeders gave themselves the same score ($p < 0.001$). Only 45.5% ($n = 33$) of raw feeders gave their veterinarian a 4 or 5 on the scale, whereas 78.0% ($n = 215$) of cooked feeders ranked their veterinarian as 4 or 5 out of 5 ($p < 0.001$) (Fig. 1).

## Free text answers

Owners provided a variety of opinions via free-text boxes when asked about perceived benefits and risks of raw feeding as well as why they chose to feed a RMBD or why they chose not to feed a RMBD to their dog. These perceptions are visually summarized in the word clouds (Figs. 2–5). The most repeated words for benefits of RMBDs were "health" (frequency ($f$) = 105), "better" ($f$ = 104), "coat" ($f$ = 59), and "teeth" ($f$ = 50) (Fig. 2) and the most repeated words for risks were "bacteria" ($f$ = 91), "nutrition" ($f$ = 72), "risk" ($f$ = 63), and "Salmonella" ($f$ = 39) (Fig. 3). When asked why they chose to feed raw, owners

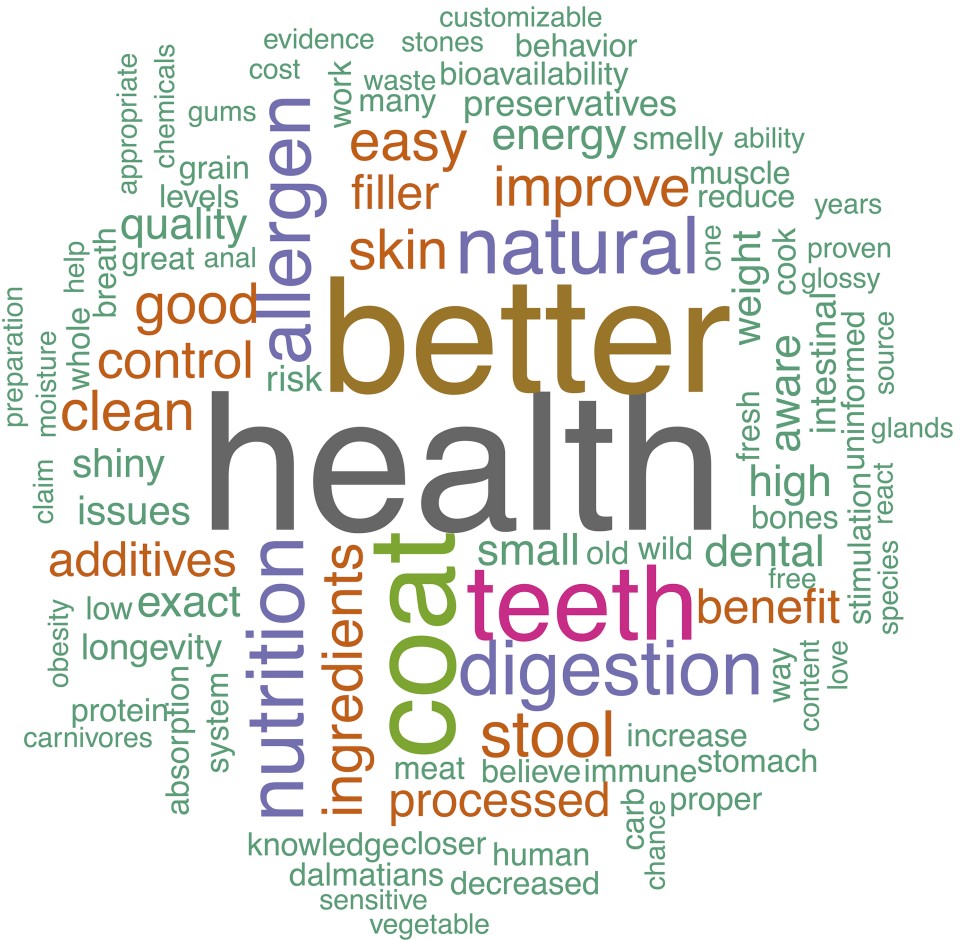

**Figure 2 Perceived benefits of raw diets.** Response to the prompt: what benefits are you aware of associated with feeding a raw diet?

most often used the words "health" ($f$ = 54), "better" ($f$ = 23), "nutrition" ($f$ = 20)" and "coat" ($f$ = 14) (Fig. 4). Conversely, when asked why they chose not to feed raw, the most common words were "expensive" ($f$ = 59), "time" ($f$ = 45), "risk" ($f$ = 42), and "convenience" ($f$ = 36) (Fig. 5).

## DISCUSSION

Our demographic data indicates that our survey respondents were overwhelmingly female ($n$ = 393, 93.6%) which seems to be a typical result in surveys of pet owners (*Morgan, Willis & Shepherd, 2017*). Though we did have responses from a breadth of countries, the United States and the United Kingdom were the two most represented countries, which is an expected result given that our questionnaire was only available in English.

Comparing raw feeders with cooked feeders revealed many key distinctions in opinion and perception between the two groups. To start, raw feeders and cooked feeders differed in their assessments of the nutritional quality of various diets. As we hypothesized, raw feeders perceived homemade and commercial RMBDs both as highly nutritious, making

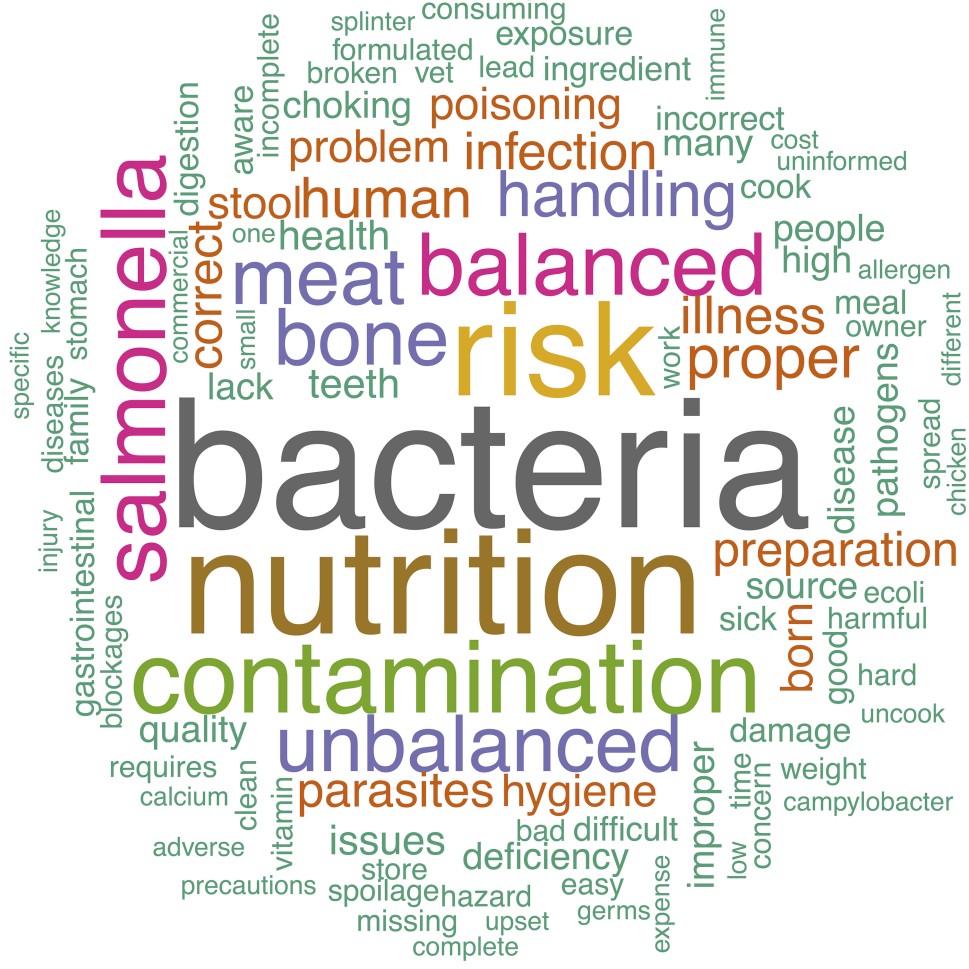

**Figure 3 Perceived risks of raw diets.** Response to the prompt: what risks are you aware of associated with feeding a raw diet?

little distinction between the two preparations. This perception is not supported by scientific evidence; numerous studies have demonstrated the risks of nutritional imbalances inherent in homemade diets (*Freeman & Michel, 2001*; *Stockman et al., 2013*; *Pedrinelli, Gomes & Carciofi, 2017*; *Dillitzer, Becker & Kienzle, 2011*). Even homemade diets formulated by veterinarians have been shown to sometimes be incomplete though they tend to have fewer and less severe deficiencies than those formulated by non-veterinarians (*Freeman & Michel, 2001*). Homemade diets should always be formulated in consultation with board-certified veterinary nutritionists to ensure they are properly balanced. A wide array of medical conditions can be caused by improper nutrient balance including nutritional secondary hyperparathyroidism, developmental orthopedic conditions, and even canine nutritional hyperthyroidism; there are multiple documented cases of dogs developing these conditions as a result of eating improperly balanced homemade RMBDs (*Taylor et al., 2009*; *Krook & Whalen, 2010*; *Zeugswetter, Vogelsinger & Handl, 2013*; *Köhler, Stengel & Neiger-Casas, 2012*). These risks are concerning, especially as a recent international study of pet owners found that 89% of raw-feeding dog

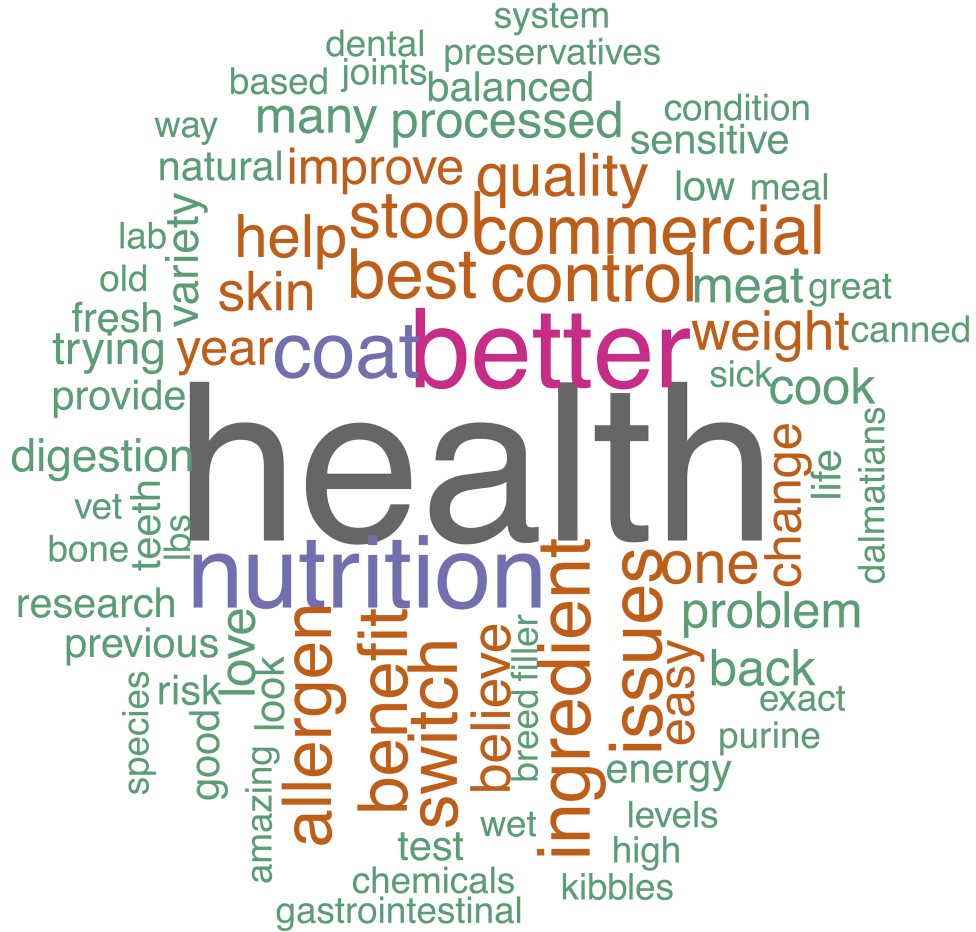

**Figure 4 Reasons for feeding a raw diet.** Response to the prompt: if you feed your dog a raw diet, why?

owners fed homemade raw diets (*Dodd et al., 2020*). A separate survey aimed specifically at raw feeders found that only 15% of respondents formulated their dog's RMBD with guidance from a veterinarian or nutritionist (*Morelli et al., 2019*). The fact that raw feeders are not making a distinction between the nutritional quality of homemade and commercial RMBDs and that many seem to be formulating their dog's diet without appropriate guidance suggests a need for further owner education on the risks of feeding an improperly balanced homemade diet. By comparison, the majority of cooked feeders did distinguish between their assessment of the two preparations, with few of them rating homemade RMBDs as highly nutritious and a larger number rating both commercial RMBDs and CCDs as highly nutritious. It is worth noting that a RMBD being commercially produced is not a guarantee that it is nutritionally balanced, particularly given that legal standards for pet food vary from country to country. The WSAVA recommends only feeding commercial pet food (raw or cooked) from companies that meet specific standards, including the full-time employment of a board-certified veterinary nutritionist (*World Small Animal Veterinary Association Global Nutrition Committee, 2013*).

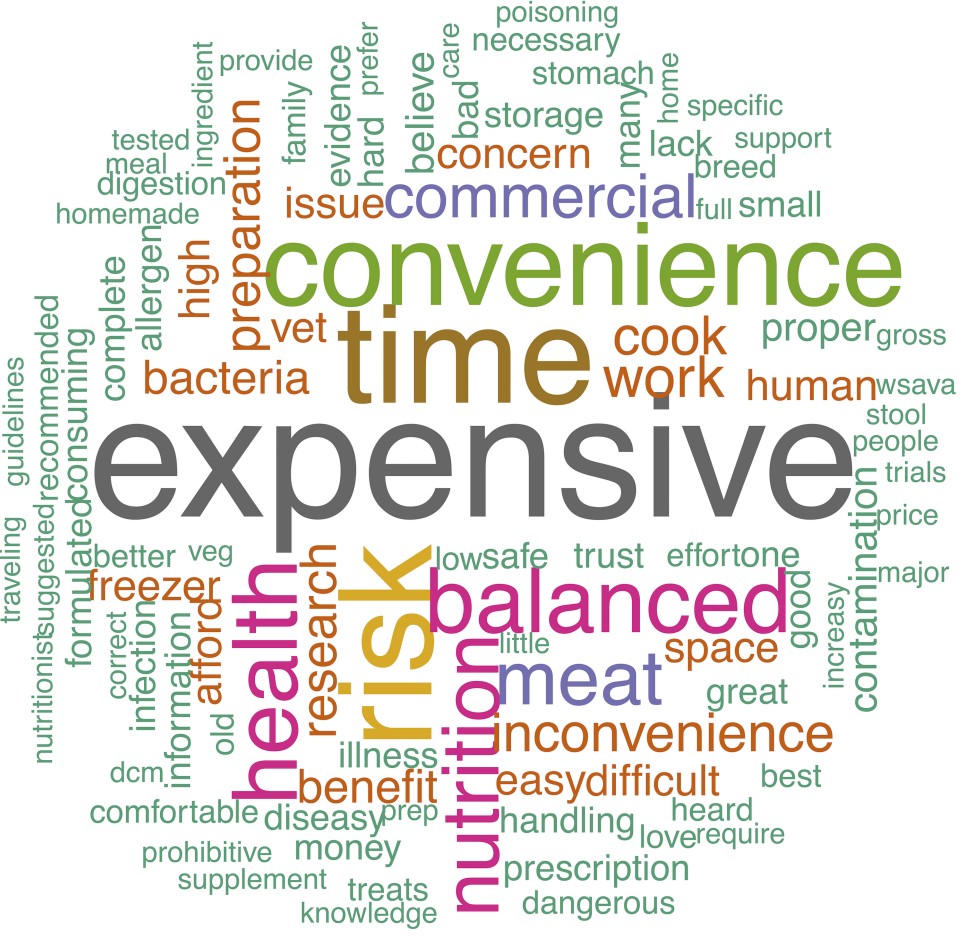

**Figure 5 Reasons for not feeding a raw diet.** Response to the prompt: if you do not feed your dog a raw diet, why not?         

Unsurprisingly only one in eight raw feeders viewed CCDs as highly nutritious. Cooked feeders demonstrated interesting perceptions of CCDs, with slightly over half of them ($n = 164$, 52.7%) rating CCDs as highly nutritious; that is an alarmingly low number considering that is the diet they chose to feed to their dogs. When we combine these findings with the top reasons that owners provided for not feeding RMBDs ("time", "expensive", "convenience"), we can postulate that diet choice is multifactorial and that lifestyle factors may be playing a larger role than nutritional value for some owners. This is an area that begs further research to understand exactly why cooked feeders are choosing to feed a diet they do not view as particularly high in nutritional value.

When it comes to risks associated with raw and cooked diets, raw feeders and cooked feeders again had different perceptions. Cooked feeders were far more likely to rate both commercial and homemade RMBDs as higher risk to both dog and human health than CCDs. This demonstrates an awareness of the published dangers surrounding the handling and consumption of raw meat products. One of those risks is of zoonotic infection with *Salmonella*, which is one of the most commonly found pathogens in RMBDs and may pose a greater threat to owners than to their pets; dogs have been found less likely

to exhibit clinical signs of *Salmonella* infections even while shedding the bacteria into their environments, and multiple studies have demonstrated higher incidence of *Salmonella* shedding in dogs fed RMBDs (*Gruenberg, 2019*; *Reimschuessel et al., 2017*; *Lefebvre et al., 2008*; *Lenz et al., 2009*). This risk is not necessarily mitigated by strict hygiene standards; *Salmonella* species have been shown to persist in dog bowls used for RMBDs, even after being cleaned in a dishwasher at 85 °C or scrubbed and soaked in bleach (*Weese & Rousseau, 2006*). Salmonella is the most common pathogenic risk from raw feeding cited by our respondents, though it is by no means the only risk associated with RMBDs; other examples of pathogens found in studies of RMBDs include the bacteria *Escherichia coli* species, *Campylobacter* species, *Clostridium perfringens*, and *Brucella suis*, as well as the parasites *Toxoplasma gondii*, *Sarcoystis cruzi*, and *Sarcoystis tenella*. These contaminants can lead to a range of disorders, including polyradiculoneuritis in the case of *Campylobacter spp.* infection (*Hellgren et al., 2019*; *Van Bree et al., 2018*; *Van Dijk et al., 2018*; *Martinez-Anton et al., 2018*). There have also been concerning reports of antimicrobial-resistant pathogens found in commercial RMBDs, from strains of *E. coli* to various *Salmonella* serovars, and there are ongoing reports of cats in the UK contracting Tuberculosis from a particular brand of commercial RMBD (*Finley et al., 2007*; *Nilsson, 2015*; *O'Halloran et al., 2019*). Many of these risks have been further explored by *Davies, Lawes & Wales (2019)* in their 2019 review. One recent study found that only 63 out of 16,475 raw-feeding households self-reported that a member of their household became ill due to raw products fed to their pet(s), and of those households, only 39 had the pathogen confirmed by a laboratory (*Anturaniemi et al., 2019*). This seems to indicate a very minimal human risk from feeding RMBDs to pets, however the two most commonly reported pathogens in that study, *Campylobacter* and *Salmonella* are both considered to be widely underdiagnosed and underreported (*Wagenaar, French & Havelaar, 2013*; *WHO, 2015*). It is difficult to gauge the exact degree to which zoonotic transmission of pathogens occurs from the feeding of RMBDs but the risk is certainly present. While responses from cooked feeders imply some level of understanding of the pathogenic risks of RMBDs, they also demonstrate a perception that commercial RMBDs are less risky than homemade RMBDs when there is little evidence to support this. By contrast, raw feeders were not likely to rate any type of RMBD as highly risky to either human or animal health. This correlates with Morelli et al who found that 65% of raw feeders believed that RMBD consumption cannot make dogs ill (*Morelli et al., 2019*). These findings clearly indicate a need for further owner education on the pathogenic risks of RMDBs.

Interestingly, a quarter of cooked feeders ($n = 45$, 25.1%) and roughly two-thirds of raw feeders ($n = 13$, 65.3%) rated CCDs as highly risky when it comes to canine health. It is worth noting that participants were not given an opportunity to explain the reasoning behind their ratings, so it is unclear what perceived risks were being associated with CCDs. Cooked pet food can be host to microbial pathogens, however, the risk is significantly lower than for RMBDs; a study of over 1,000 commercially available pet foods found evidence of *Salmonella* or *Listeria* species in 40.8% of raw samples and only 0.42% of cooked ones (*Nemser et al., 2014*). However there have been some highly publicized incidents of dry food recalls including a problem in 2007 with melamine contamination

and more recently excess levels of vitamin D which may have been a factor for owners in answering this question (*U.S. Food and Drug Administration, 2018*, *2020*). Exploring owner-perceived risks associated with commercially cooked food would be an interesting area for further research.

When respondents were asked to rate their veterinarian's knowledge of nutrition, 59.9% (*n* = 249) of owners surveyed believed their veterinarian to be highly knowledgeable (4 or 5 out of 5). A prior survey by Morgan et al found that only 35.9% of surveyed dog owners trusted their veterinarians as a knowledge resource with respect to pet nutrition (*Morgan, Willis & Shepherd, 2017*). While that statistic makes our figure seem surprisingly high, further analysis of our data shows a large difference in the assessment of veterinarian nutritional knowledge depending on the diet fed by the owner in question; cooked feeders gave their veterinarian an average score of 3.9 out of 5 while raw feeders gave an average score of only 2.9 out of 5. The stark difference between the perceptions of the two groups demonstrates a need for better communication between veterinarians and their raw feeding clients, with specific regard to veterinarians' ability to inspire confidence in their own nutritional knowledge. Furthermore, this emphasizes the need for a strong foundation in nutrition in veterinary education alongside the communication skills and confidence to raise questions and start evidence-based discussions about nutrition in the consulting room.

One of the novel aspects of this research was allowing owners to provide their opinions on raw feeding in their own words using free-text boxes. When all owners were asked about the benefits of raw feeding, specific keywords repeatedly arose such as "teeth", "coat", "natural", "digestion" and "allergen", along with general positive words like "better" and "health". Highlighting the areas where owners think raw feeding benefits their dogs can serve as a starting point for veterinarians to open discussion about nutrition and help them to understand what owners find appealing about the practice of feeding RMBDs.

When asked why raw feeders chose to feed RMBDs, frequently repeated words were more often vague: "health", "better", "issues" and "benefits". These did not further the understanding of the motivation behind raw feeding and may expose an inability of raw feeders to agree on reasons to choose RMBDs. The lack of specific terms also emphasizes the dearth of peer-reviewed published data on the benefits of raw feeding.

Asking owners about the risks associated with raw feeding generated less ambiguous results, with words like "bacteria", "Salmonella", "contamination", "bone" and "unbalanced" being frequently repeated. This suggests that at least a subset of the population is informed about some of the documented risks of RMBDs. When owners who did not feed RMBDs were asked why they avoid the practice, they used tangible words again, with "time", "expensive", "risk", and "convenience" being some of the most repeated words. It is worth pointing out that the most common answers for why owners chose not to feed RMBDs do not directly correlate with the risks they listed previously. This perhaps indicates that it is not the risks associated with RMBDs that is preventing these owners from choosing that diet for their dog.

Compiling the free-text data into word clouds or tag clouds provides an intuitive and accessible way to visualize the data, rather than simply relying on frequency

graphs. It demonstrates the complexity and diversity of data while highlighting the most important and frequently used keywords as the largest and centrally located words (*Bateman, Gutwin & Nacenta, 2008*). The use of word clouds originated in the world wide web as a way to compile tags but has since evolved as a useful tool for text analytics (*Heimerl et al., 2014*). Their use in scientific literature has not been widespread, thus this application represents an innovative way to present our data. Our word clouds summarize a massive amount of free-text data and allow us to pick out trends and keywords quickly and efficiently while providing an esthetically pleasing medium that is easy for a layperson to interpret.

The free-text boxes were a particular strength of this survey as it allowed owners a chance to express their perceptions in their own words. Additional strengths included our wide geographic spread of participants and variety in professional background of participants. Limitations of the survey include aspects that potentially restricted participation, in particular the need for respondents to be proficient in the English language as well as have internet and social media access. The varying geographical and cultural backgrounds of respondents may also have led to some confusion in terminology, for example, the term "traditional" was used multiple times in the survey and yet will imply different connotations to a respondent depending on their background. As with many studies, sample size and participant engagement could have been more robust and will be a point of attention for future research.

## CONCLUSIONS

This study used a novel questionnaire to assess dog owner perceptions around canine feeding of raw meat-based and commercial cooked diets. Dog owners have immense choice when deciding on a diet for their pets and the data indicated that an increasing number of them are choosing RMBDs, despite concerns cited by leading veterinary bodies (*Davies, Lawes & Wales, 2019*). This contradiction indicates a clear need to understand what specifically is driving owners to choose RMBDs or CCDs, though it is equally important to assess what causes owners to avoid feeding these diet types too.

We found our hypotheses to be largely correct; dog owners who choose to feed RMBDs generally viewed the practice to be less risky to both human and dog health than owners who do not feed RMBDs. Raw feeders also rated raw diets as significantly more nutritious than CCDs. Moreover, over six in seven raw feeders perceived themselves as highly knowledgeable about nutrition, while only half viewed their veterinarian as knowledgeable. Conversely, cooked feeders perceived their veterinarians as more knowledgeable than they were about their dog's nutrition, but only half of them viewed CCDs as a nutritious diet. Further potential areas of research could include probing into specific claims made by raw feeders in the free-text portion of the questionnaire as well as exploring why cooked feeders choose their diet.

## ACKNOWLEDGEMENTS

We would like to acknowledge Tim Parkin and William Houston for their help with statistics and coding.

### Funding
This work was supported by MSD Animal Health. The funders had no role in study design, data collection and analysis, decision to publish, or preparation of the manuscript.

### Grant Disclosures
The following grant information was disclosed by the authors:
MSD Animal Health.

### Competing Interests
The authors declare that they have no competing interests.

### Author Contributions

- Alysia Empert-Gallegos conceived and designed the experiments, performed the experiments, analyzed the data, prepared figures and/or tables, authored or reviewed drafts of the paper, and approved the final draft.
- Sally Hill conceived and designed the experiments, performed the experiments, analyzed the data, prepared figures and/or tables, authored or reviewed drafts of the paper, and approved the final draft.
- Philippa S. Yam conceived and designed the experiments, authored or reviewed drafts of the paper, supervised and approved the final draft.

### Human Ethics
The following information was supplied relating to ethical approvals (i.e., approving body and any reference numbers):

The University of Glasgow College of Medical, Veterinary & Life Sciences Ethics Committee for Non-Clinical Research Involving Human Participants granted ethical approval to carry out this survey (Application Ref: 200180125) and informed consent was obtained for all participants.

### Data Availability
Data is available at Figshare: Empert-Gallegos, Alysia; Poole, Sarah (2020): RRaw Questionnaire Raw Data.xlsx. figshare. Dataset. DOI 10.6084/m9.figshare.12404432.v1.

### Supplemental Information
Supplemental information for this article can be found online at http://dx.doi.org/10.7717/peerj.10383#supplemental-information.

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
