# Peer review of "Insights into dog owner perspectives on risks, benefits, and nutritional value of raw diets compared to commercial cooked diets"

_PeerJ, doi:10.7717/peerj.10383_

## Round 0.1 · original submission · Minor Revisions

It is my pleasure to handle your article. This is a very interesting article that will fill the gap of knowledge on this file. However, to complete the scientific paper. I would like you to revision this manuscript according to our reviewer's comments. I am waiting for the revised version.

·

Basic reporting

Language is excellent.
Some remarkable research papers in the field were not cited and discussed.
The article is well structured; however, the discussion section does not follow the results section, and this creates come confusion. Some figures are redundant. No copy of the survey was provided, thus making it difficult to fully understand results source.
The submission is self-contained.

Experimental design

Further details could be provided according to CHERRIES guidelines for e-surveys.

Validity of the findings

This study is interesting, provides new data in a little explored field, innovative use of word clouds for displaying data.
The presentation of data sometimes is too shallow, and some precise numbers are missing.
Conclusions are ok.

Additional comments

TITLE
I think “responsible raw?” is misleading and should be removed from the title. Responsibility could not be deducted from this survey.

INTRODUCTION
L57 and throughout: Seen the recent history of pet food, “traditional” is a term that should be avoided when speaking of canine nutrition.

M&M
Further details about the survey creation may be reported following the CHERRIES guidelines on e-surveys.

RESULTS
- L93-100: Please give precise numbers about the owners demographics
- Was the signalment of owners’ dogs asked?
- Even when “most” and “majority” terms are used, please report the precise n/% in brackets
- Please consider writing all results using both numbers and percentages (n/N, n%)
- please be consistent using one decimal after comma
- please verify that percentages sum is 100%; if needed, report the % of “other” option whenever available to reach 100%

DISCUSSION
- Please respect the same topics order in the results section
- L202-206: balancing canine diets is not “difficult”, it should be emphasized that this should be done by board-certified veterinary nutritionists whose consultation should always be recommended; usually you do not write the name of universities when citing papers in the main text
- L226-247: it may be pointed out that interviewed owners were mainly concerned about Salmonella (probably because it is the most talked-about bacterium in general?), even if many studies showed that RMBDs are contaminated by many pathogens
- There are two recent studies that deserve to be added and widely discussed: Raw meat-based diets for dogs: survey of owners’ motivations, attitudes and practices (Morelli et al., 2019); Owners’ perception of acquiring infections through raw pet food: a comprehensive internet-based survey (Anturaniemi et al., 2019)
- Commercial RMBDs were shown to be even more contaminated that human grade meat and byproducts (Zoonotic bacteria and parasites found in raw meat-based diets for cats
and dogs; Van Bree et al. 2018; Evaluation of microbial contamination and effects of storage in raw meat‐based dog foods purchased online, Tuberculosis due to Mycobacterium bovis in pet cats associated with feeding a commercial raw food diet, O’Halloran et al. 2019; Morelli et al. 2020); I would add some discussion about safety perception in home-prepared RMBDs vs commercial RMBDs

TABLES
Such tables and values are not effective in displaying Likert-type scores. Please choose a more appropriate style or graph

FIGURES
Figures 5 and 6 seem redundant to me as they show the same results of figures 1 and 2. I would suggest removing them

OTHER
Please provide a copy of the survey to help me understand the source of the results.

·

Basic reporting

This article is well written, clear, concise, and easy to follow. References to contemporary researched background information are appropriate.

Within the results, please include a table of respondent demographics. As the information is currently presented, the reader is unable to determine what other options were available to the respondent other than female, omnivorous, and not working as part of the animal industry. Additionally, which animal industry is being referred to? Please provide more information here. Please also include the number of participants from each country.

While reporting the proportion of respondents ranking 4 to 5 out of 5 in the different categories does convey appropriate information in the results, please include in the tables the prevalence of each rank, not just 4 to 5 out of 5.

The use of the word clouds is a relatively novel way to report concepts identified by the respondents. However, there are some improvements required for these to be more precise. For example, in Figure one, there are multiple concepts that have been reported separately, such as "poop", "poops", "stools" and "allergy", "alergens". Please review the data and combine these similar words into single words so that their impact can be more accurately evaluated. In Figure two, there appear to be concepts that have been separated and are not being depicted clearly. For example, "getting" is unlikely to be a risk that was reported, and this data should be reviewed and the world cloud presented in a manner in which the entire concept is displayed. These recommendations extend to all the word cloud figures.

In figure 6 there is a mistake in the title and description as these do not match. The title is the most commonly cited RISKS but in the description it includes the prompt BENEFITS. Figure 7 has a similar mistake. Figure 8 the prompt does not make sense: "if you feed your dog a raw diet, why not" - likely it was intended to read "if you DO NOT feed your dog a raw diet, why not".

Experimental design

The article presents original primary research with a well defined research question. The background clearly demonstrates the relevance and meaningfulness of the study.

The methodology is vague and does not describe the study with adequate detail to replicate. It is unclear how data regarding dog diet was collected as described in the results section. Within the section "Establishment of Diet", the 'main diet' of the dogs represented in the study is described, but there is no description of how this information was obtained. Please include a more detailed description of how this information was obtained in the methods section and clearly indicate what "predominantly" and "main diet" refer to within the context of the dog diets.

Validity of the findings

The conclusions reported by the authors are consistent with the results. The findings presented within this article will be of practical value to veterinary practitioners. Some novel inferences were made and identify areas for further research.

Most of the data is reported as frequency and proportion, for which no statistical tests are required. However, when comparisons between RMBD and CCD feeders were made, statistical differences were determined using chi-squared testing. This test may not be the most appropriate for ranked outcomes (Tables 1 - 3) and a different statistical test should be considered.

In the discussion it is mentioned that raw feeders do not make a distinction between nutritional quality of homemade and commercial RMBD (Lines 210-211). Please include a brief comment regarding the nutritional quality of commercial RMBD, as there have been studies indicating that commercial RMBD may fare no better than homemade RMBD in terms of nutritional content and balance. This may vary geographically as regulations controlling pet food manufacture and sales practices differ.

Additional comments

This is an interesting article that presents some new insights and reinforces previous findings regarding the behaviours surrounding feeding RMBD to dogs. In general the article is well written, there are just some minor revisions required particularly regarding the methods and results.

·

Basic reporting

The figure labels don’t appear to be listed.
Within lines 33 and 156, one of the most repeated words for risks has been reported as “bacteria(1)”. Relevance of the number 1 here is unclear and needs to be clarified.

Experimental design

Methods are clearly described, using sufficient information to enable reproduction by another investigator, however it would have been valuable for a copy of the questionnaire to have been supplied by the authors. It would also be useful to know how many Facebook pages/groups the questionnaire was posted to initially.

Validity of the findings

Impact of the study has been assessed, with findings clearly reported and predicted to be of interest and value to a wide variety of stakeholders. Further detail regarding the number of survey participants hailing from the USA and UK versus the other countries would have been interesting to include.

Additional comments

This is an insightful and informative article that demonstrates originality, helps to fill the acknowledged knowledge gap in this field and is a pleasure to read.

---

## Round 0.2 · Minor Revisions

Please address the remaining minor issues.

·

Basic reporting

no comment

Experimental design

no comment

Validity of the findings

no comment

Additional comments

Dear authors, it was a pleasure for me to revise your manuscript, which has greatly improved after the revisions. I think your work deserves to be published in PeerJ, I just leave a few final comments. Nice job, best regards!

THROUGHOUT THE TEXT
Please be consistent using one decimal after comma: Sorry if this was not clear (I try my best but I am not native speaker), I meant that it would be appropriate to 1) write just one decimal number (e.g. 93.57% becomes 93.6%), as this is commonly enough for surveys and you do not need more overly precise results 2) report the same quantity of numbers of decimals (e.g. 20% becomes 20.0%)

INTRODUCTION
L44: Google with capital letter

M&M
L80: “section 11 consisted of a thank you”, is it too informal? Sorry again, I am not native speaker, it just looks not very elegant to me. Please consider rephrasing if deemed appropriate

RESULTS
L108-109: “One entry consisted only of the consent and no other data so it was not included in the analysis of results.” This sentence serves no purpose and can be removed

DISCUSSION
L189-190: “for reasons that remain unclear” I do not think this is a “scientific mystery” (probably women take more care of the family meals, pets included) and I would avoid this phrase, but feel free to disagree with my opinion

FIGURES
Figures 1: For each parameter, the upper line is the judgement of cooked feeders and the lower line is that of raw feeders? Because it is difficult to understand from the box on the right (whose colors are just black and grey which I cannot find in the figure). Please put a clearer graph legend about owners or just specify this in the figure title/description

·

Basic reporting

No comment

Experimental design

Description of the experimental design has been greatly improved from the first submission.

Validity of the findings

No comment

Additional comments

My comments were addressed appropriately.

·

Basic reporting

The article is well written with clear and unambiguous use of English language throughout. Text is technically correct and the article conforms to professional standards of courtesy and expression.

Sufficient introduction and background provided, demonstrating how the work fits into the broader field of knowledge. Additional current and relevant literature has been incorporated and cited correctly, providing sufficient context.

The revised figures and tables are relevant to the content of the article and are of sufficient resolution. The addition of a table outlining details of respondent demographics is helpful. The use of a diverging stacked bar chart in figure 1 is an acceptable method for presenting rating scale data, however the layout could be improved. The percentage figures are hard to read and identifying which results relate to cooked versus raw feeders could be made more explicit.

Experimental design

The study is within the aims and scope of Peer J and the research objectives are well defined, relevant and meaningful.

Methods are clearly described. Additional information regarding the experimental design and survey details has been supplied, together with a copy of the survey, enabling reproduction by another investigator.

Validity of the findings

Impact of the study has been assessed and additional details have now been added, providing a more comprehensive overview and clearer reporting of the findings.

In general, results have been reported using either (n=X, X%) or X% (n=X). Please adopt this same format throughout the manuscript and ensure consistency; the results reported in tracked lines 327 and 328 (untracked lines 268 and 269) differ. Please also provide both numbers and percentages. The percentage is provided but the corresponding number is missing from tracked lines 210 and 283, however this detail has been included in the untracked version. There is an n missing from tracked line 113 (untracked line 111) - currently 306, 73.21% instead of n=306, 73.21%.

Additional comments

Thank you for re-submitting your manuscript. The revisions have enhanced your article which is predicted to be of great interest and value to readers of PeerJ, however comments have been provided for your consideration in relation to figure 1 and the overall reporting of results.

Your survey has generated a substantial amount of data, some of which has not been reported or fully explored in this manuscript and will hopefully result in the production of further publications in the future.

---

## Round 0.3 · accepted · Accept

This manuscript has greatly improved after the revisions. I think your work deserves to be published in PeerJ. Congratulations.